# Discovery of a Natural Syk Inhibitor from Chinese Medicine through a Docking-Based Virtual Screening and Biological Assay Study

**DOI:** 10.3390/molecules23123114

**Published:** 2018-11-28

**Authors:** Xing Wang, Junfang Guo, Zhongqi Ning, Xia Wu

**Affiliations:** 1School of Traditional Chinese Medicine, Capital Medical University, Fengtai District, Beijing 100069, China; kingstar1016@sina.com (X.W.); fanggj1519@ccmu.edu.cn (J.G.); ningzhongqi@163.com (Z.N.); 2Beijing Key Lab of Traditional Chinese Medicine Collateral Disease Theory Research, Capital Medical University, Fengtai District, Beijing 100069, China

**Keywords:** Syk inhibitors, tanshinone I, virtual screening, Syk inhibition assay, cell-based assay

## Abstract

Spleen tyrosine kinase (Syk) is a critical target protein for treating immunoreceptor signalling-mediated allergies. In this study, a virtual screening of an in-house Chinese medicine database followed by biological assays was carried out to identify novel Syk inhibitors. A molecular docking method was employed to screen for compounds with potential Syk inhibitory activity. Then, an in vitro kinase inhibition assay was performed to verify the Syk inhibitory activity of the virtual screening hits. Subsequently, a β-hexosaminidase release assay was conducted to evaluate the anti-mast cell degranulation activity of the active compounds. Finally, tanshinone I was confirmed as a Syk inhibitor (IC_50_ = 1.64 μM) and exhibited anti-mast cell degranulation activity in vitro (IC_50_ = 2.76 μM). Docking studies showed that Pro455, Gln462, Leu377, and Lys458 were key amino acid residues for Syk inhibitory activity. This study demonstrated that tanshinone I is a Syk inhibitor with mast cell degranulation inhibitory activity. Tanshinone I may be a potential lead compound for developing effective and safe Syk-inhibiting drugs.

## 1. Introduction

Spleen tyrosine kinase (Syk) is a 72 kDa cytosolic non-receptor tyrosine kinase that is involved in signal transduction by the classic immunoreceptors, including mast cells, B-lymphocytes, and macrophages, and contributes to diverse cellular responses, including histamine release, cell proliferation, differentiation, and phagocytosis [1]. Studies have shown that when stimulated through FcεRI, Syk-deficient mast cells cannot synthesize leukotrienes, secrete cytokines, or degranulate [2]. Inhibition of Syk-mediated immunoreceptor (B-cell receptors, T-cell receptors and Fc receptors) signalling leads to inhibition of mast cells, macrophages and B-cell activation and subsequent release of inflammatory modulators [3]. Recently, Buyanravjikh et al. reported that cryptotanshinone could inhibit IgE-mediated mast cell degranulation through theinhibition of tyrosine kinase-dependent degranulation signalling pathways involving Syk and Lyn [4]. Therefore, seeking safe Syk small molecule inhibitors has attracted considerable interest in a number of therapeutic areas, including the treatment of allergic asthma, rhinitis, rheumatoid arthritis and chronic lymphocytic leukaemia [5,6,7]. However, there is currently no Syk small molecule inhibitor drug on the market, although some potential compounds have been tested in clinical trials [8,9].

Traditional Chinese medicine (TCM), with an over 2000 year history in the treatment of diseases, is a valuable source for drug leads and candidates in the process of drug discovery. Many popular natural compounds, such as artemisinin, berberine, ginkgolide, and paclitaxel, extracted from TCMs, play an important role in disease treatments [10,11,12,13]. In this work, a combined approach involving docking-based virtual screening and bioassay-based validation was performed to identify natural Syk small molecule inhibitors from TCMs. A schematic diagram of bioactive compound discovery in this study is shown in Figure 1. The top hits in molecular docking were tested for Syk inhibition using a kinase inhibition assay. Additionally, a degranulation assay was carried out in RBL-2H3 cells to evaluate the bioactivity of compounds at the cellular level. Finally, tanshinone I (CAS: 568-73-0) was identified as a Syk inhibitor and had anti-mast cell degranulation activity in vitro. Our findings help to further reveal more biological activities of tanshinone I as a Syk inhibitor. Additionally, the binding mode between tanshinone I and Syk provides a reference for structural optimization and activity enhancement. The approach used in this work is feasible for identifying Syk inhibitors from TCMs and beneficial to the development of effective and safe Syk-inhibiting drugs.

## 2. Results and Discussion

### 2.1. Docking-Based Virtual Screening Targeting Syk

Docking-based virtual screening was employed in this work to identify potential Syk inhibitors. The co-crystal structure of Syk (PDB ID: 4PUZ) in complex with a known inhibitor, CG9 (molecular formula: C_23_H_21_N_7_O), named 6-(1*H*-indazol-6-yl)-*N*-[4-(morpholin-4-yl)phenyl]imidazo[1,2-a]pyrazin-8-amine, was obtained from a protein data bank (https://www.rcsb.org/) as a template for docking screening. An in-house database containing 320 natural compounds was initially filtered by PAINS (http://www.cbligand.org/PAINS/login.php) and Lipinski’s ‘rule of five’ to remove potential false positive molecules [14], which resulted in a list of 251 compounds. To validate the docking programme, co-crystallized CG9 was re-docked into the active site of Syk and the docked conformation with the highest total score of 6.07 was selected as the most likely binding conformation. As shown in Figure 2, the root-mean-square deviation (RMSD) of 0.52 Å between the co-crystallized and the docked conformations of CG9 indicated a high reliability of the docking programme in reproducing the experimentally observed binding mode. The results showed that the docked conformation was nearly in the same position as the co-crystallized conformation. Ala451, Asp512, Leu377, Gly454, and Leu501 were key amino acids for Syk inhibition in the CG9 binding site, which is consistent with previous reports in the literature [15]. A total of 251 natural compounds were successively docked into the active site of Syk, resulting in a hit list of 18 compounds with docking scores above 6.00 (Table 1). The 18 hits were further evaluated for their in vitro inhibition activity against Syk.

### 2.2. Tanshinone I Dose-Dependently Inhibited Syk Activity

The 18 virtual hits from the docking screening were assessed for their Syk-inhibitory activity using an ATP determination kit named ADP-Glo^TM^ kinase assay kit (Promega, Madison, WI, USA) according to themanufacturer’s instructions. Staurosporine, a known Syk inhibitor, was used as a positive control [16]. A kinase buffer containing 5% DMSO was used as a vehicle control. The inhibition of test compounds at a concentration of 30 μM was measured, and compounds with inhibition rates of >60% were selected for IC_50_ value determination. Among the 18 compounds, only tanshinone I significantly inhibited Syk compared to the vehicle control (*p* < 0.05) (Figure 3). A further dose-effect analysis found that tanshinone I could dose-dependently inhibittheSyk activity with an IC_50_ of 1.64 μM (Figure 4).

### 2.3. Tanshinone I Dose-Dependently Inhibited Mast Cell Degranulation

To evaluate the anti-mast cell degranulation activity of tanshinone I, the release rate of β-hexosaminidase, an important biomarker in degranulation, was measured in RBL-2H3 cells after antigen stimulation. Chloroquine, a known mast cell degranulation inhibitor, was used as a positive control [17]. As shown in Figure 5A, chloroquine (positive control) and 2.22–60.00 micromoles of tanshinone I significantly inhibited β-hexosaminidase release in IgE/BSA-stimulated RBL-2H3 cells. The half-inhibitory concentration for the inhibition of Syk by tanshinone I was determined to be 2.76 μM (Figure 5B). All experiments at each concentration of tanshinone I had three replicates and were repeated three times.

### 2.4. Binding Site of Tanshinone I in Syk Model

Most of the known Syk inhibitor molecules have specific structural scaffolds, such as pyridine-2-carboxamide, pyrazin-8-amine, pyrimidine-8-carboxamide, pyrimidin-4-one, pyridazine-3-carboxamide, pyrimidine-5-carboxamide, (3*E*)-3-(1*H*-pyrrol-2-ylmethylidene)-1*H*-indol-2-one and methylquinoline-4,6-diamine [15,18,19,20,21,22,23,24,25]. Tanshinone I, a phenanthrene derivative, has a simpler scaffold than those of known inhibitors. A ligand-Syk docking model was performed to predict the molecular mechanism of tanshinone I recognition and binding to Syk. As shown in Figure 6, the keto group and furan ring of tanshinone I were inserted into the catalytic centre of Syk to form hydrogen bond interactions with the side chains of Gln462 and Lys458, respectively. Moreover, the aromatic rings were surrounded by hydrophobic residues (Leu377 and Pro455), suggesting that hydrophobic interactions occurred between tanshinone I and Syk. The pharmacophore model characterizing ligand-protein interactions was consist of three aromatic rings and two hydrogen bond receptors, which were potential key structural features of tanshinone I for Syk inhibition.

Tanshinone I is an important lipophilic diterpene extracted from *Radix Salvia miltiorrhizae* (Danshen), a well-known traditional herbal medicine in China that has a variety of pharmacological effects, including antioxidant, anti-inflammatory, heart-protective, and anti-osteoporotic effects [26,27]. Studies have found that tanshinones have anti-inflammatory, anti-allergic, and other pharmacological effects [28,29]. Choiet al. reported that tanshinones possibly exert their anti-allergic activities by affecting FcεRI-mediated tyrosine phosphorylation of ERK and PLCγ2 [30]. Buyanravjikh et al. reported that cryptotanshinone, a natural compound extracted from *Salvia miltiorrhiza* Bunge, had an inhibitory effect on IgE/antigen-mediated mast cell degranulation through the inhibition of tyrosine kinase-dependent degranulation signalling pathways [4]. This study demonstrates, for the first time, that tanshinone I is a direct Syk inhibitor and has anti-mast cell degranulation activity in vitro, which may provide a perspective for elucidating the molecular mechanism of tanshinone I for its anti-allergic and other pharmacological effects.

To further evaluate the reliability of our VS workflow, a retrospective assessment was carried out [31]. As shown in the Appendix A, simpler ligand-based approaches such as fingerprint similarity search and 3D pharmacophore model screening showed a low potency in identifying Tanshinone I from the natural compound database. Virtual screening based on Surflex-Dock not only increases the probability of identifying active compounds targeting Syk, but also predicts the interaction between the bioactive molecule and target protein.

## 3. Materials and Methods

### 3.1. Molecular Docking

Molecular docking was conducted using the Surflex-Dock module in the SYBYL-X 1.3 software (Tripos, Inc., St. Louis, MO, USA) [32,33,34,35]. All 320 molecules from our in-house natural compound database were downloaded from the PubChem database (https://pubchem.ncbi.nlm.nih.gov/) in mol2 format. All hydrogen atoms were added, and the partial atomic charges of the atoms of each compound were assigned using the Gasteiger-Hückel method. Each structure was energy-minimized using the Tripos force field with a distance-dependent dielectric constant and the Powell conjugate gradient algorithm convergence criteria, which partially accounts for the shielding effects of the aqueous environment on electrostatic interactions [36]. These conformations were used as starting conformations to perform molecular docking. The crystal structure of Syk (PDB ID: 4PUZ), determined by X-ray diffraction at a 2.09 Å resolution, was chosen as a docking protein model [37]. All co-crystallized water molecules of the protein model were removed, and polar hydrogen atoms were added using SYBYL X-1.3. The protein model was assigned a force field using Gasteiger-Marsili charges and then energy-optimized for 1000 iterations using the default parameters in SYBYL X-1.3. The amino acid residues within 0.5 Å around the CG9 ligand were defined as a docking pocket using a ProtoMol-based method [38]. The ProtoMol in Surflex-Dock utilized various molecular fragments, such as CH_4_, C=O, and N-H, to represent hydrophobic groups, hydrogen bond donors and hydrogen bond acceptors, respectively. The remaining parameters for docking were used on the default settings.

To validate the performance of the docking procedure, the co-crystallized CG9 ligand was extracted and re-docked into the active sites of the model. The root-mean-square deviation (RMSD) between the re-docking and co-crystallized conformation of CG9 was calculated according to the following formula [39,40]:(1)$$\mathrm{RMSD}=\sqrt{\frac{1}{N}\overset{i=N}{\underset{i=1}{\sum }}d_{i}^{2}}$$
where *d* is the distance between *N* pairs of equivalent atoms excluding hydrogen. A lower RMSD indicates a greater overlap between the re-docked and co-crystallized conformation of theCG9 ligand. The compounds were docked into the active pocket of Syk, resulting in a hit list with a total score for each molecule. A higher total score indicates a higher binding force between the ligand and protein model. A cut-off value of six was used to select a reasonable number of virtual hits for further investigation [41].

### 3.2. In Vitro Kinase Inhibition Assays

To verify the Syk inhibitory activity of the test compounds, a luminescent kinase assay was performed to measure the ADP produced in a kinase reaction utilizing an ADP-Glo^TM^ kinase assay kit (Promega, Madison, WI, USA) following the manufacturer’s instructions. In this assay, the ADP produced by the Syk activity was converted to ATP, which is the substrate of luciferase, consequently leading to the production of light. Therefore, the luminescent signal directly correlates with the kinase inhibitory activity of the test compounds. Staurosporine (CAS No. 62996-74-1) was used as a positive control for the Syk activity assay. A kinase buffer containing 5% DMSO was used as a negative control. All test compounds were purchased from the National Institutes for Food and Drug Control (Beijing, China), and the purity of each compound was greater than 98% based on HPLC analysis.

All reactions were performed in white, non-binding 384-well plates with flat bottoms (Corning #3824, Corning Glass Works, Corning, NY, USA). The kinase reaction was performed in Tris buffer (40 mM, pH 7.5) containing 20 mM MgCl_2_, 0.1 mg/mL bovine serum albumin and 50.0 μM DTT. Four microliters of Syk (ab60886) solution (1 ng/μL in kinase buffer) and 2.0 μL of test compound solution (150 μM in kinase buffer containing 5% DMSO) were incubated for 15 min at 27 °C. Then, 4.0 μL of substrate solution (0.2 μg/μL polypeptide (4:1 Glu, Tyr) (Abcam catalogue no. ab204877, Cambridge, UK) in a kinase buffer containing 10.0 μM ATP) was added to activate the kinase reaction for 1 hour at 27 °C. The kinase reaction was terminated by adding 5 μL ADP-Glo™ reagent into 5.0 μL Syk reaction solution. The mixture was incubated for 40 min to deplete the unconsumed ATP. Finally, 10.0 μL of kinase detection reagent was added, and the mixture was incubated for 1 h to convert ADP to ATP; then, luciferase and luciferin were added to detect ATP. The luminescence was determined using an Envision 2104 Multilabel Reader (Perkin-Elmer, Eden Prairie, MN, USA). To avoid false negative results caused by the self-luminescence of compounds, the luminescence values of 300 compounds were detected. The fluorescence values were quantified in relative fluorescence units (RFU).

### 3.3. Cell Culture

RBL-2H3 cells, a rat cell line that is useful for the in vitro screening for anti-allergic agents [42,43], were obtained from the American Type Culture Collection (ATCC; Manassas, VA, USA) and grown in minimal essential medium (Gibco, Life Technologies, Carlsbad, CA, USA) supplemented with 10% (*v*/*v*) heat-inactivated foetal bovine serum, 100 U/ml penicillin, and 100 μg/mL streptomycin (Gibco BRL, Grand Island, NY, USA). The cells were cultured in an incubator at 37 °C under humidified 5% CO_2_.

### 3.4. β-Hexosaminidase Release Assay

The assay was performed as previously described [44] with some modifications. The measurement of released β-hexosaminidase from RBL-2H3 cells was used as an indicator of mast cell degranulation. *P*-nitrophenyl-*N*-acetyl-β-*O*-glucosamine (PNAG) could be hydrolysed to *p*-nitrophenol by β-hexosaminidase at 37 °C, pH 4.5. *P*-nitrophenol could be quantified by spectrophotometrically measuring the ionized product after adding a stop buffer (pH 10). RBL-2H3 cells were seeded into 96-well plates at a density of 2 × 10^4^ cells/well and incubated for 12 h at 37 °C. After washing the cells with PBS, the medium was changed to phenol red-free Dulbecco’s-modified Eagle’s medium (Grand Island, NY, USA). The cells were then sensitized with anti-DNP monoclonal IgE (Sigma D-8406, St. Louis, MO, USA) at 100 ng/mL and incubated for 24 h. One hour before the antigen challenge, the test group cells were cultured by adding 20.0 μL of different concentrations of tanshinone I in 0.25% DMSO. The negative control group, positive control group, and induction group cells were cultured by adding an equal volume of HBSS and 0.25% DMSO, chloroquine solution and phenol red-free DMEM, respectively. The cells were then stimulated with 100 ng/mL DNP-HSA (Sigma-Aldrich, St. Louis, MO, USA) for 30 min at 37 °C and placed in an ice bath at 0 °C for 10 min. Fifty microliters of the supernatant or total lysate were mixed with substrate solution (1 mM PNAG in 0.1 M citric acid buffer, pH 4.5) and incubated for 90 min at 37 °C. The reaction was stopped by adding 200 μL of stop buffer (0.1 M NaHCO_3_/Na_2_CO_3_, pH 10). The absorbance of each well was measured at 405 nm (OD) with a FlexStation 3 microplate reader (Molecular Devices, Sunnyvale, CA, USA). Background fluorescence of the substrate solution alone was subtracted from all readings. The percentage of β-hexosaminidase release was calculated by using the following equation:(2)$$\beta \mathrm{-hexosaminidase}\;\mathrm{release}\;(\%)\;=\;\frac{{\mathrm{OD}}_{supernatant}\;-{\;\mathrm{OD}}_{blank}}{{\mathrm{OD}}_{lysate}\;-{\;\mathrm{OD}}_{blank}}\times 100$$

### 3.5. Statistics

Data are reported as the mean ± standard error of the mean (SEM). Significant differences between the groups were analysed using one-way analysis of variance (ANOVA). The experimental data were analysed by the GraphPad Prism software (version 5, GraphPad Software Inc., San Diego, CA, USA). For all tests, *p* < 0.05 was considered statistically significant.

## 4. Conclusions

In this study, we carried out a virtual screening of an in-house Chinese medicine database to identify a novel scaffold for Syk inhibitors. Structure-based virtual screening and biological experiments were conducted to further verify the biological activity of the compounds and investigate the binding site between ligands and target proteins. The results presented here led us to conclude that tanshinone I is an active inhibitor against Syk. It is suggested to perform some subsequent point mutations to prove the contribution of amino acid residues to Syk inhibitory activity. Moreover, in-depth biochemical experiments including Syk signalling assay and a kinase selectivity profiling test could be performed to evaluate the mechanism of tanshinone I exhibiting anti-mast cell degranulation activity. Additionally, binding studies could be conducted to evaluate the kinetic characteristics between tanshinone I and Sykin the future. In short, tanshinone I with its unique scaffold can be used as a starting point for lead optimization to develop effective and safe Syk-inhibiting drugs.

## Figures and Tables

**Figure 1 molecules-23-03114-f001:**
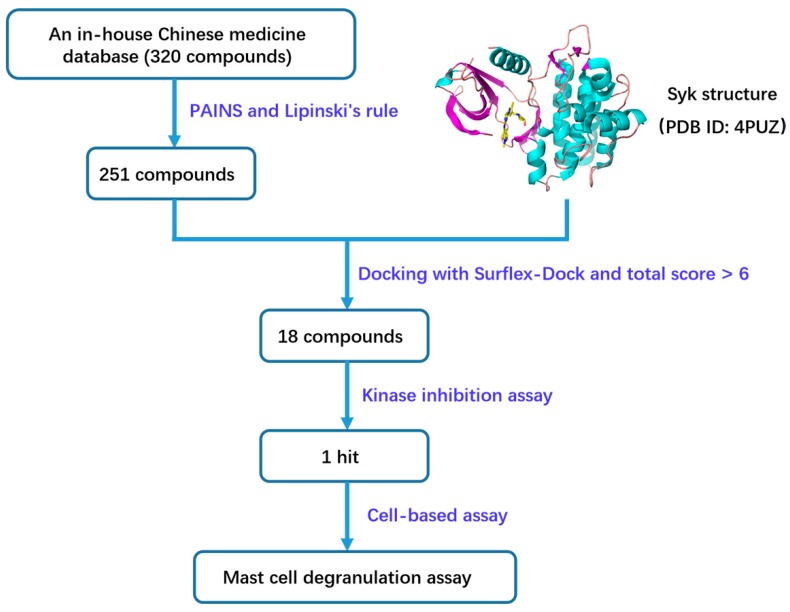
The schematic diagram of active compound discovery based on virtual screening and biological assays.

**Figure 2 molecules-23-03114-f002:**
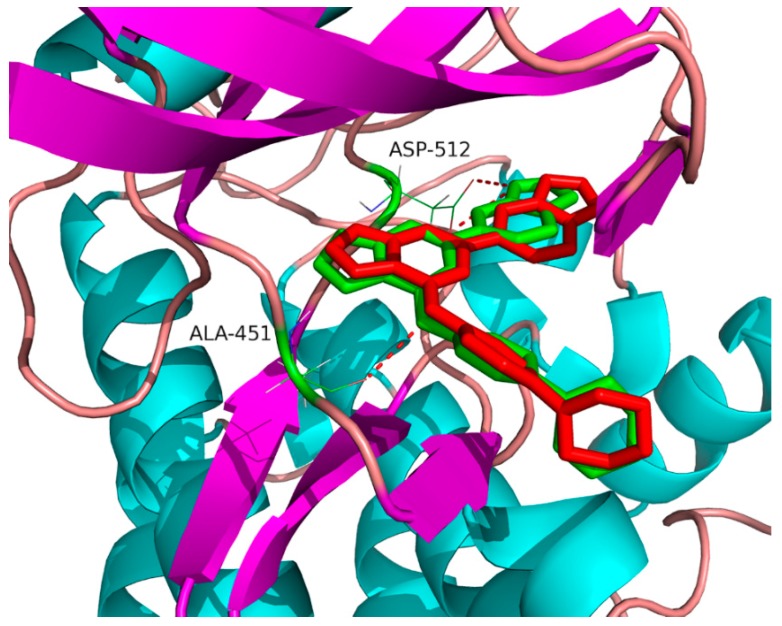
The binding site of theCG9 ligand in the inhibitor binding site of Syk. The red and green compounds represent the co-crystal and docked conformations of theCG9 ligand in Surflex-docking, respectively. The red dotted lines indicate hydrogen bond interactions between CG9 and Syk.

**Figure 3 molecules-23-03114-f003:**
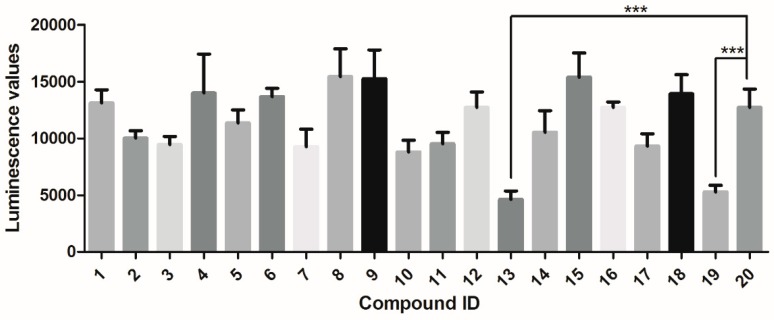
The luminescence values of the Syk solution after incubation with 18 test compounds in ADP-Glo^TM^ kinase assays. The luminescence value was detected in the presence of 1 ng/μL Syk incubated with 18 compounds (30 μM in the total reaction system) using an ADP-Glo^TM^ kinase assay kit for primary screening. Information about compounds 1 to 18 can be found in Table 1. Compounds **19** and **20** represent thepositive and negative control, respectively. The error bars indicate the standard error (SE) of three replicates. *** means *p* < 0.001.

**Figure 4 molecules-23-03114-f004:**
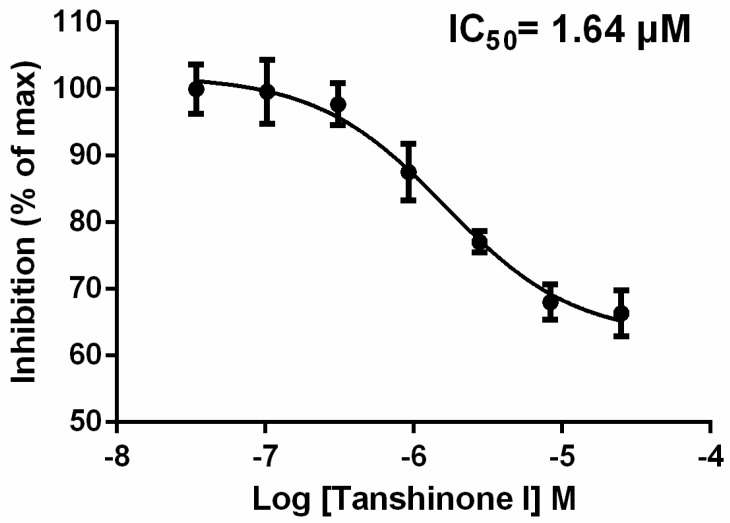
The dose-response curve of tanshinone I inhibition of Syk activity. All error bars represent the SE of three replicates.

**Figure 5 molecules-23-03114-f005:**
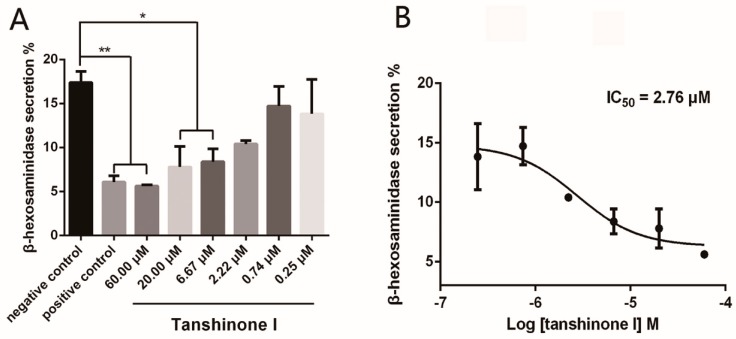
The inhibition of Syk activity by different concentrations of tanshinone I (**A**) and dose-response curve analysis (**B**). All error bars represent the SE of thethree replicates. ** means *p* < 0.01 and * means *p* < 0.05.

**Figure 6 molecules-23-03114-f006:**
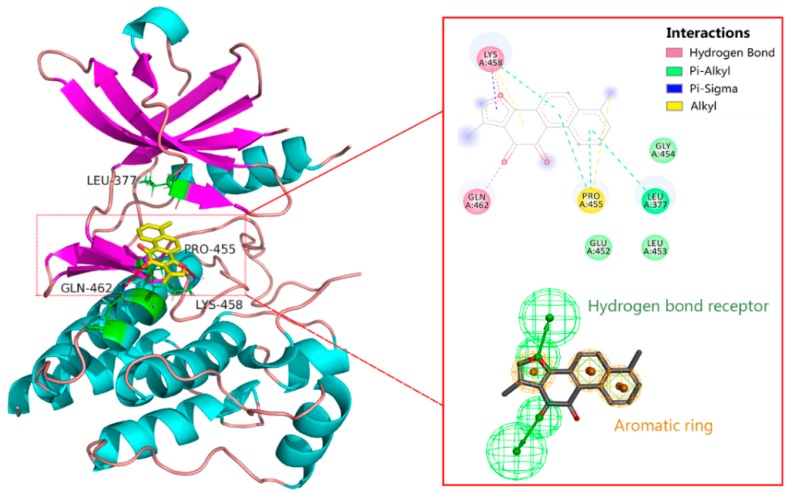
The binding site of tanshinone I in the inhibitor binding site of Syk.

**Table 1 molecules-23-03114-t001:** The hits from docking-based virtual screening.

ID	Total Score	Name	CAS No.	Sources
1	9.61	10-Gingerol	23513-15-7	*Zingiber officinale*
2	8.02	Calceolarioside B	105471-98-5	*Akebia quinata*
3	7.71	6-Gingerol	23513-14-6	*Zingiber officinale*
4	7.59	Corylifolinin	20784-50-3	*Glycyrrhiza uralensis*
5	7.48	Arbutin	497-76-7	*Saussurea lappa*
6	7.37	Piceid	27208-80-6	*Lilium brownii var. viridulum*
7	7.07	Ostruthin	148-83-4	*Angelica decursiva*
8	7.01	Wogonoside	51059-44-0	*Scutellaria baicalensis*
9	6.91	Mulberroside A	102841-42-9	*Morus alba*
10	6.88	Anisodamine hydrobromide	55449-49-5	*Anisodustanguticus (Maxim.) Pascher*
11	6.84	Curcumin	458-37-7	*Curcuma zedoaria*
12	6.55	Dehydrodiisoeugenol	2680-81-1	*Syzygium aromaticum*
13	6.30	Tanshinone I	568-73-0	*Salvia miltiorrhiza*
14	6.25	l-Arctigenin	7770-78-7	*Arctium lappa*
15	6.21	Mulberroside C	102841-43-0	*Morus alba*
16	6.18	6-Shogaol	555-66-8	*Zingiber officinale*
17	6.15	Propyl gallate	121-79-9	*Rheum officinale*
18	6.03	Sesamolin	526-07-8	*Sesamum indicum*

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
