# Peer review of "Discovery of a Natural Syk Inhibitor from Chinese Medicine through a Docking-Based Virtual Screening and Biological Assay Study"

_molecules, 2018, doi:10.3390/molecules23123114_

Round 1
Reviewer 1 Report
The authors make the discovery of a Syk inhibitor from Chinese Medicine
through a virtual screening using computational techniques. Out of 251
compounds selected with drug like properties they find 18 promisng
candidates. Enzymatic evaluation shows that only one of the cmpouns
inhibits the enzyme at the low micromolar range. They finally evaluate
this compound in cellular experiments of anti-mast cell degranulation.
The manuscript is well written and I only found small mistakes that have
been pointed out.
Line 84, I would remove the parenthesis, later in he text the concentration is stated.
line 97, there is no need to include all the names here since they are already in table 1. Just keep what 19 and 20 are.
Author Response
Response to Reviewer 1 Comments
Point 1: Line 84, I would remove the parenthesis, later in he text the concentration is stated.
Response 1: Thank you for your recommendation. We have removed the parenthesis since the sample information can be found in method section 3.2.
Point 2: line 97, there is no need to include all the names here since they are already in table 1. Just keep what 19 and 20 are.
Response 2: Thank you for your comments. We have removed the names of compounds 1 to 18 since this information can be found in table 1.
Reviewer 2 Report
The paper by Wu et al. describes the virtual screening of an in-house Chinese medicine database followed by biological assays carried out to identify novel Syk inhibitors. This study demonstrated that tanshinone I is a Syk inhibitor with mast cell degranulation inhibitory activity.
The experimental data are quite sound and the discussion and well organized. The results obtained are interesting, but some issues need to be addressed before the manuscript can be considered for publication:
1) Introduction: The following reference should be added and discussed by the authors: “Cryptotanshinone inhibits IgE‑mediated degranulation through inhibition of spleen tyrosine kinase and tyrosine‑protein kinase phosphorylation in mast cells”, by Sumiyasuren Buyanravjikh et al., 2018, https://doi.org/10.3892/mmr.2018.9042.
2) Authors should describe the features of the pharmacophore model created on the basis of the docking analysis.
3) Authors can perform molecular dynamics simulations, to validate the docking analysis. Eventually MD simulations can be performed with CG9 and inactive compounds to provide more confidence on the predicted binding mode of tanshinone I.
4) In order to assess the solidity of the VS workflow, a retrospective evaluation can be carried out, such as fingerprint similarity searches or 3D ligand similarity screenings (see i.e. L.R. Chiarelli, et al. Eur. J. Med. Chem. 2018, 155, 754-763. DOI: 10.1016/j.ejmech.2018.06.033).
5) Authors can deepen the biochemical study, investigating the mechanism of action of tanshinone I (i.e. determining the Ki with Syk; see S Braselmann, 2006, J. Pharmacol. Exp. Therapeutics, 319, 998, https://www.ncbi.nlm.nih.gov/pubmed/16946104) and determining its selectivity against a kinase screening panel (i.e. Kinase Profiler panel, Millipore).
6) The References should be standardized to the ACS style guide.
Author Response
Response to Reviewer 2 Comments
Point 1: Introduction: The following reference should be added and discussed by the authors: “Cryptotanshinone inhibits IgE‑mediated degranulation through inhibition of spleen tyrosine kinase and tyrosine‑protein kinase phosphorylation in mast cells”, by Sumiyasuren Buyanravjikh et al., 2018, https://doi.org/10.3892/mmr.2018.9042.
Response 1: Thank you sincerely for your recommendation. This important reference has been added and discussed in the revised manuscript.
Point 2: Authors should describe the features of the pharmacophore model created on the basis of the docking analysis.
Response 2: Thank you for your recommendation. Based on the docking analysis, we have analysed the pharmacophore model that characterize ligand-protein interactions, which help to reveal the key structural features of tanshinone I for Syk inhibition.
Point 3: Authors can perform molecular dynamics simulations, to validate the docking analysis. Eventually MD simulations can be performed with CG9 and inactive compounds to provide more confidence on the predicted binding mode of tanshinone I.
Response 3: Thank you sincerely for your recommendation. However, due to the limitations in our laboratory, we are currently unable to conduct molecular dynamics in this study. Your suggestion is so valuable and we will carry out MD simulations in the future to improve our work.
Point 4: In order to assess the solidity of the VS workflow, a retrospective evaluation can be carried out, such as fingerprint similarity searches or 3D ligand similarity screenings (see i.e. L.R. Chiarelli, et al. Eur. J. Med. Chem. 2018, 155, 754-763. DOI: 10.1016/j.ejmech.2018.06.033).
Response 4: Thank you sincerely for your recommendation. We have carried out a retrospective evaluation using fingerprint similarity searches and 3D pharmacophore model methods. But they all showed low potency in identifying Tanshinone I from the natural compound database. Studies have shown that virtual screening based on molecular docking is efficient. Detailed data and discussions could be found in supplementary materials. Thank you for your comments and it makes our methodological discussions more fulfilling.
Point 5: Authors can deepen the biochemical study, investigating the mechanism of action of tanshinone I (i.e. determining the Ki with Syk; see S Braselmann, 2006, J. Pharmacol. Exp. Therapeutics, 319, 998, https://www.ncbi.nlm.nih.gov/pubmed/16946104) and determining its selectivity against a kinase screening panel (i.e. Kinase Profiler panel, Millipore).
Response 5: Thank you for your comments sincerely. However, due to the limitations of our laboratory, we cannot carry out Ki determination or kinase selectivity analysis currently. The relevant work prospects have been proposed and discussed in the revised manuscript. Your comments provide a very valuable guide for our future study. And our laboratory will be dedicated to improve our study according to your suggestions.
The focus of this study was to identify tanshinone I as a syk inhibitor and to evaluate its inhibition of mast cell degranulation activity in vitro. In addition, the binding characteristics between tanshinone I and Syk were predicted. We don't know whether the current work meets the publication requirements of MOLECULES.
Point 6: The References should be standardized to the ACS style guide.
Response 6: Thank you for your comments. We have standardized all references to ACS style. At the same time, we have checked the format of the references according to the journal requirements.
Reviewer 3 Report
The manuscript titled ‘Discovery of a Natural Syk Inhibitor from Chinese Medicine through a Docking-based Virtual Screening and Biological Assay Study’ presents a study of the identification of Syk inhibitors through molecular docking and biological assays. Tanshinone I was found to be an active compound inhibiting Syk in an enzymatic assay. The experiments are reported clearly and concisely and the results are discussed appropriately. Overall, the manuscript is well-written. However, there are certain concerns which should be addressed before the manuscript can be considered for acceptance.
- Figure 5A has some errors in representation. Positive control shows maximum secretion whereas negative control shows minimum secretion. Moreover, figure 5B shows IC50 of 2.76 µM while the text in section 2.3 shows IC50 as 3.52 μM.
- It has been reported that the mast cell degranulation inhibition can be also due to off-target effects of tested compounds. To prove that the degranulation effect is through Syk inhibition, Syk signaling assay should be performed (Pharmacological Research 99 (2015) 116–124).
- Also, kinase selectivity profiling should be performed to determine selective Syk inhibition.
- Provide the rationale for performing mast cell degranulation assay.
Author Response
Point 1: Figure 5A has some errors in representation. Positive control shows maximum secretion whereas negative control shows minimum secretion. Moreover, figure 5B shows IC50 of 2.76 µM while the text in section 2.3 shows IC50 as 3.52 μM.
Response 1: Thank you for your careful review sincerely. We have carefully checked these representation and the errors has been corrected in the revised manuscript.
Point 2: It has been reported that the mast cell degranulation inhibition can be also due to off-target effects of tested compounds. To prove that the degranulation effect is through Syk inhibition, Syk signaling assay should be performed (Pharmacological Research 99 (2015) 116–124).
Response 2: Thank you for your comments sincerely. However, due to limitations of our laboratory, we cannot carry out Syk signaling assay currently. The relevant work prospects have been proposed in the revised manuscript. Your comments provide a very valuable guide for our future study. And we will further study the molecular mechanism of tanshinone I inhibiting mast cell degranulation in the future work according to your suggestions.
This manuscript focuses on the fact that tanshinone I was identified as a Syk inhibitor that showing anti-mast cell degranulation activity in vitro. In addition, the binding characteristics between tanshinone I and Syk were predicted. We want to confirm whether the current work meets the publication requirements of MOLECULES.
Point 3: Also, kinase selectivity profiling should be performed to determine selective Syk inhibition.
Response 3: Thank you for your comments sincerely. Due to limitations in laboratory conditions and research funding, we cannot carry out kinase selectivity profiling currently. The relevant work prospects have been proposed in the revised manuscript. And we will be dedicated to improve our study according to your suggestions.
Point 4: Provide the rationale for performing mast cell degranulation assay.
Response 4: Thank you for your comments. We have added the rationale for performing mast cell degranulation assay in the revised manuscript.
The measurement of released β-hexosaminidase from RBL-2H3 cells was used as an indicator of mast cell degranulation. P-nitrophenyl-N-acetyl-β-O-glucosamine (PNAG) could be hydrolysed to p-nitrophenol by β-hexosaminidase at 37 °C, pH 4.5. P-nitrophenol could be quantified by spectrophotometrically measuring the ionized product after adding stop buffer (pH 10). So the degree of degranulation of mast cells can be determined by measuring the hydrolytic degree of PNAG under the catalysis of β-hexosaminidase from mast cells.
Round 2
Reviewer 2 Report
The Authors fully answered to the queries, so the manuscript can be accepted in the present form.
Author Response
Response to Reviewer 2 Comments
Point 1: The Authors fully answered to the queries, so the manuscript can be accepted in the present form.
Response 1: Thank you sincerely for your comments, which provide a very valuable guide for our future study.
Reviewer 3 Report
As indicated before, Figure 5A has an error.
Negative control inhibits degranulation potently as compared to Chloroquine which was used as the positive control according to Figure 5A. Please check if the representation is correct in the graph.
Author Response
Response to Reviewer 3 Comments
Point 1: As indicated before, Figure 5A has an error.
Response 1: Thank you for your careful review sincerely. We are sorry that we forgot to correct this error in Figure 5A. We have corrected it in the latest version, please check it. Thank you again for your comments.